# Dual tRNA mimicry in the Cricket Paralysis Virus IRES uncovers an unexpected similarity with the Hepatitis C Virus IRES

Vera P Pisareva[1], Andrey V Pisarev[1]*, Israel S Fernández[2]*

[1]Department of Cell Biology, SUNY Downstate Medical Center, Brooklyn, United States; [2]Department of Biochemistry and Molecular Biophysics, Columbia University, New York, United States

**Abstract** Co-opting the cellular machinery for protein production is a compulsory requirement for viruses. The Cricket Paralysis Virus employs an Internal Ribosomal Entry Site (CrPV-IRES) to express its structural genes in the late stage of infection. Ribosome hijacking is achieved by a sophisticated use of molecular mimicry to tRNA and mRNA, employed to manipulate intrinsically dynamic components of the ribosome. Binding and translocation through the ribosome is required for this IRES to initiate translation. We report two structures, solved by single particle electron cryo-microscopy (cryoEM), of a double translocated CrPV-IRES with aminoacyl-tRNA in the peptidyl site (P site) of the ribosome. CrPV-IRES adopts a previously unseen conformation, mimicking the acceptor stem of a canonical E site tRNA. The structures suggest a mechanism for the positioning of the first aminoacyl-tRNA shared with the distantly related Hepatitis C Virus IRES.
DOI: https://doi.org/10.7554/eLife.34062.001

*For correspondence:
andrey.pisarev@downstate.edu (AVP);
isf2106@cumc.columbia.edu (ISF)

**Competing interests:** The authors declare that no competing interests exist.

## Introduction

Translation initiation is the most complex and highly regulated step of protein synthesis (*Schmeing and Ramakrishnan, 2009*). Canonical initiation results in the formation of an elongation-competent ribosome with an aminoacyl-tRNA base paired with messenger RNA (mRNA) at the peptidyl site (P site) of the ribosome. Translation initiation in eukaryotes is achieved by a highly sophisticated mechanism (*Jackson et al., 2010*). Most eukaryotic mRNAs contain a unique nucleotide structure at their 5' end, known as the cap structure. The multi-subunit initiation factor eIF4F binds the cap structure and recruits the 43S complex consisting of the small ribosomal subunit (40S), eIF2/GTP/Met-tRNA$^{Met}$ ternary complex, eIF3, eIF1, eIF1A, and eIF5 (*Jackson et al., 2010*). The attached 43S complex scans the 5'-untranslated region of the mRNA downstream to the initiation codon, where it forms the 48S initiation complex with the established codon-anticodon base-pairing in the ribosomal P site. Finally, eIF5B, the eukaryotic ortholog of the bacterial initiation factor 2 (IF2), promotes the recruitment of the large ribosomal subunit (60S) and the formation of the elongation-competent 80S ribosome (*Fernández et al., 2013*).

Eukaryotic viruses have evolved refined molecular strategies to interfere with canonical initiation factors, leading to the hijacking of host ribosomes to produce viral proteins (*Jackson et al., 2010; Hertz and Thompson, 2011*). A common strategy used by different types of viruses relies on structured RNA sequences at the ends of their mRNAs (*Filbin and Kieft, 2009*).These sequences are called Internal Ribosomal Entry Sites (IRES) and form specific three-dimensional structures able to manipulate and co-opt the host translational machinery (*Yamamoto et al., 2017*).

**eLife digest** Viruses cannot replicate themselves, but instead depend on components of the host cell for their own survival. Once a virus successfully enters a cell, it must use part of the cell's machinery – specifically the ribosomes – to produce its own proteins. Ribosomes normally make the cell's proteins by reading instructions written in molecules known as messenger RNAs (or mRNAs for short). Viruses hijack ribosomes using structured RNA segments in its mRNAs that can mimic natural components of the cell's protein-producing machinery. These RNA sequences, known as IRESs, feature a refined balance between rigidity and flexibility. Their flexible nature has made them difficult to study in the past, though the latest advances in electron cryo-microscopy mean that IRESs can now be directly observed in complex with ribosomes.

Pisareva et al. sought to image a prototypical IRES sequence from the Cricket Paralysis Virus as it is transitioned through the ribosome. The idea was to characterize the late stages of ribosome hijacking. First, all the essential components were purified, mixed in the laboratory, and then imaged via electron cryo-microscopy. Image processing and sorting algorithms were then used to visualize the process at a high level of detail. Unexpectedly, this showed that the IRES changes shape dramatically to mimic part of another RNA molecule, a tRNA, when it reaches the so-called exit site of the ribosome. Short for transfer RNAs, tRNAs are molecules that bring the building blocks of proteins (called amino acids) to the ribosome, ready to be linked together. The shape change in the IRES is coupled with the placement of the first amino acid-loaded tRNA in a site on the ribosome that commits it to producing the viral protein.

These results illustrate the remarkable ability of RNA molecules, in general, and IRES sequences, in particular, to adopt distinctive and context-specific shapes. These features seem to be widely conserved among diverse virus families as a similar shape change has been see in the IRES of the distantly related Hepatits C Virus. Together these new insights could lead to new strategies to interfere with viral replication and further studies that deepen our understanding of how ribosome and RNA-based mechanisms work generally inside cells.

DOI: https://doi.org/10.7554/eLife.34062.002

IRES sequences are classified according to the subset of factors they require for initiation (*Filbin and Kieft, 2009*). Type IV IRES sequences, including the Cricket Paralysis Virus IRES (CrPV-IRES) and the Taura Syndrome Virus IRES (TSV-IRES) do not require initiation factors and are the best studied IRESs. Biochemical and structural studies have provided a detailed view on how these approximately 200-nucleotide-long sequences interact with and manipulate the ribosome (*Wilson et al., 2000*; *Spahn et al., 2004*; *Petrov et al., 2016*).

A modular architecture of three pseudoknots (PKI, PKII and PKIII, *Figure 1A*) is crucial for these IRESs to establish a balance between structural flexibility and rigidity, essential for interaction with the ribosome and with two elongation factors (eEF2 and eEF1A) required for IRES translocation through the ribosome (*Jan et al., 2003*). PKI mimics an anti-codon stem loop (ASL) of a tRNA interacting with its cognate mRNA codon, and plays an essential role in setting up the correct reading frame in the aminoacyl site (A site) of the ribosome (*Costantino et al., 2008*).

A ribosome primed with a type IV IRES alternates between rotated and non-rotated configurations of the small ribosomal subunit (*Fernández et al., 2014*; *Koh et al., 2014*). Similarly, after peptidyl-transfer in canonical translocation, the ribosome alternates between rotated and non-rotated configurations of the small ribosomal subunit with respect to the large ribosomal subunit (*Voorhees and Ramakrishnan, 2013*). This pre-translocation stage of the ribosome is recognized by a protein translocation factor (EF-G in Bacteria, eEF2 in Eukarya), which in GTP-bound form induces an additional rotation of the small subunit and blocks the A site of the ribosome (*Tourigny et al., 2013*). Translocation proceeds forward by a back rotation of the small subunit to recover a canonical configuration of the ribosome. This is accomplished by a swiveling movement of the head of the small subunit in an orthogonal direction respect to that of the rotation of the small subunit (*Ratje et al., 2010*). The back-rotation of the small subunit accompanied by the swiveling of the head is performed while EF-G/eEF2 are still bound (*Ramrath et al., 2013*). Once the translocation factor EF-G/eEF2 leaves the ribosome, the head of the small subunit returns to its non-

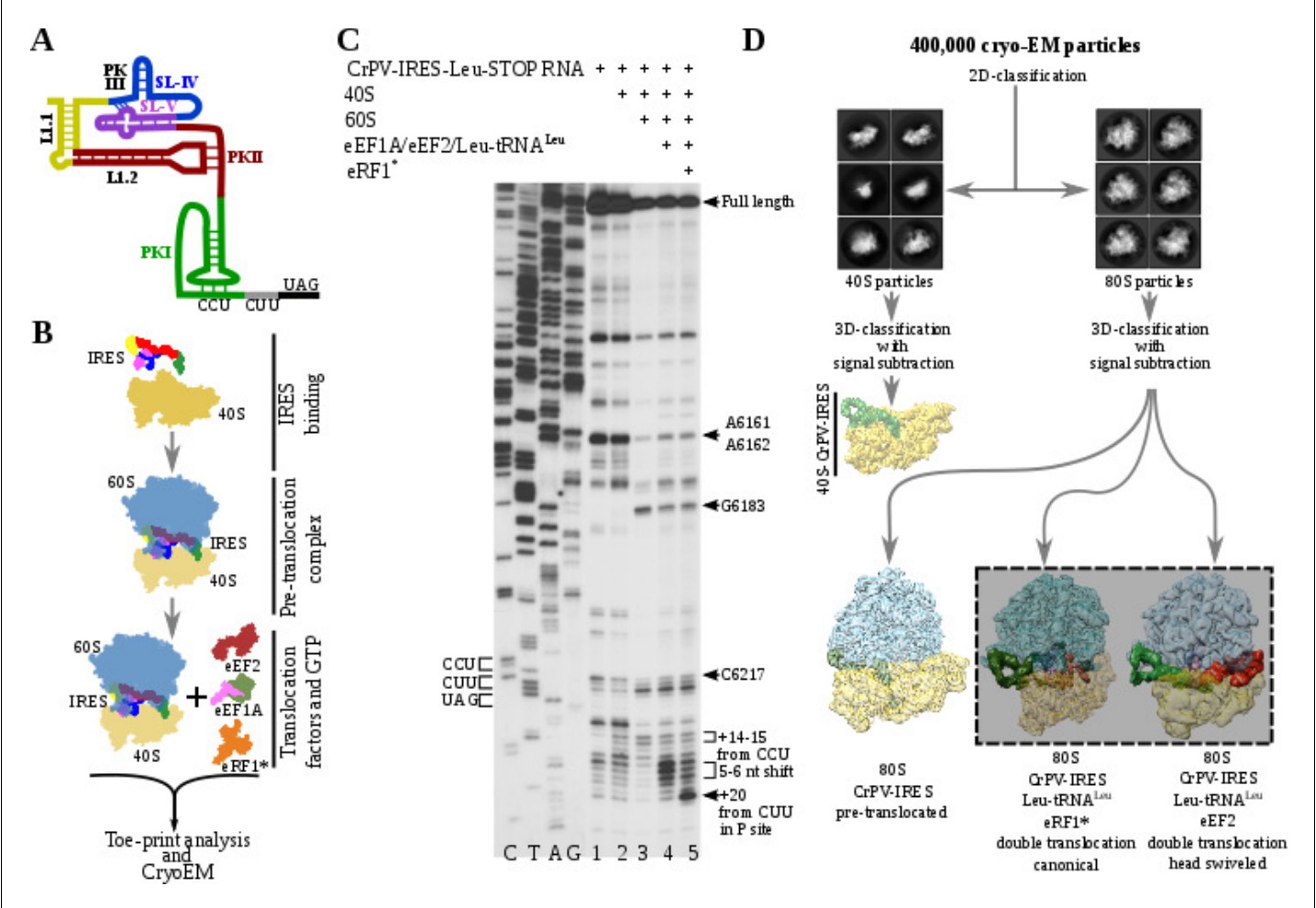

**Figure 1.** Experimental set up and cryoEM image processing workflow. (**A**) Secondary structure scheme of CPV-IRES highlighting the modular architecture consisting of three pseudoknots (PKI, PKII and PKIII). (**B**) Diagram showing the in vitro reaction set up with purified components used for the toe-printing assays as well as for cryoEM. (**C**) Toe-print analysis of ribosomal complexes assembled with components indicated on top. Toe-prints corresponding to the pre-translocated complex are labeled as +14–15 nt from CCU. Toe-prints corresponding to the double translocated complex without and with eRF1* are marked as 5-6nt shift and +20 nt from CUU, respectively. Additional toe-prints previously attributed to different CrPV-IRES/ribosome contacts are seen, in agreement with previous reports (*Muhs et al., 2015*; *Pestova et al., 2004*). (**D**) CryoEM data processing workflow employed to resolve the high compositional and conformational heterogeneity of the *in vitro* reconstituted complexes described in B. Four populations were resolved and refined to high resolution, including two exhibiting clear density for a double translocated CrPV-IRES (squared).
DOI: https://doi.org/10.7554/eLife.34062.003

swiveled configuration, rendering a ribosome primed with a peptidyl-tRNA in the P site, a deacylated tRNA in the E site and a vacant A site ready to accept the next aminoacyl-tRNA (*Noller et al., 2017*). The L1 stalk, a component of the large ribosomal subunit, also contribute to the vectorial movement of tRNAs, offering additional anchoring points to the leaving deacylated-tRNA in the E site (*Fei et al., 2008*). The movements of the L1 stalk are coordinated with those of the small subunit. Type IV IRESs require two translocation events to place the first aminoacyl-tRNA in the P site making use of the intrinsic dynamic elements of the ribosome involved in canonical translocation (*Murray et al., 2016*; *Abeyrathne et al., 2016*). However, less is known about the second translocation event, required for the first aminoacyl-tRNA to enter the P site, ending the unusual initiation pathway followed by this type of IRESs.

We report the visualization by means of single-particle electron cryo-microscopy (cryoEM) of two related states of the mammalian ribosome with a double translocated CrPV-IRES and P site aminoacyl-tRNA at 3.2 and 4.75 Ångstroms resolution. The head swiveling of the small ribosomal subunit

plays a fundamental role in the late step of this translocation event, inducing a remarkable conformational change on the PKI of the CrPV-IRES, which becomes disassembled, to mimic the acceptor stem of a E site tRNA.

## Results

To analyze the integrity and stability of complexes for cryoEM, we assembled ribosomal complexes with a double translocated CrPV-IRES in a mammalian reconstituted system from individual components in the presence of GTP (*Figure 1B and C*).The translocation efficiency was monitored by toe-printing. A pre-translocation complex, assembled by mixing of CrPV-IRES with 40S and 60S subunits, results in a +14–15 nt toe-print signal from the CCU (sequence belonging to the PKI) and thus indicate the presence of PKI in the ribosomal A site (*Figure 1C*, lane 3). The addition of elongation factors leads to a 5–6 nucleotides toe-print shift showing the double translocation event (*Figure 1C*, lane 4). However, the similar intensity of several bands, with difference in one nucleotide, suggests instability of the double translocated IRES or frame ambiguity. We reasoned this ambiguity could be explained by the absence of an A site ligand, what would allow partial back translocation of the IRES even in the presence of a translocated aminoacyl-tRNA in the P site. Similarly, in the single translocated CrPV-IRES cryoEM reconstruction, it was necessary the addition of an A site ligand (*Muhs et al., 2015*). In this report, the mutation of the first sense codon to a stop codon and the addition of the release factor 1 (eRF1) were applied to prevent spontaneous back-translocation of the IRES. Thus, to stabilize a double translocated complex, we mutated the second sense codon to a UAG stop codon (*Figure 1A*).The supplementation of the reaction with a mutated and catalytically inactive version of the release factor 1 (eRF1*, AGQ mutation) causes a +20 nt toe-print signal from the first sense codon (leucine CUU codon) in the P site, which is in a good agreement with previous report (*Muhs et al., 2015*), indicating proper binding of eRF1* (*Figure 1C*, lane 5). The simultaneous decrease of intensity for the 5–6 nucleotides toe-print suggests a more homogeneous complex, suitable for structural studies. Given the conformation of the P site tRNA is completely compatible with that of a translating ribosome, we believe the addition of eRF1* in the present sample does not significantly affect the conformation of the CrPV-IRES.

Maximum likelihood particle sorting methods implemented in RELION (*Scheres, 2012*) were applied to a large cryoEM dataset at two different stages (*Figure 1D*). An initial classification in two dimensions allowed for the separation of full ribosome (80S) from small subunit (40S) particles. The two sorted subgroups were further classified using masking methods with signal subtraction with focus in the inter-subunit space, where this type of IRES binds as well as canonical translation factors. The L1 stalk was also included in the masked area to allow for a wider sampling. This strategy revealed, in a single classification step, several sub-populations, reflecting the heterogeneity of the sample. A binary 80S/CrPV-IRES complex in a pre-translocation conformation could be identified as well as two sub-populations with CrPV-IRES in a double translocated state. In the double translocated reconstructions, the ribosome adopts a non-rotated configuration of the small subunit, with clear density for aminoacyl-tRNA in the P site and a either eEF2 or eRF1* in the A site (*Figure 1D*, *Figure 2—figure supplement 1*, *Figure 3—figure supplement 1* and *Table 1*).

Recent studies by single-molecule FRET (smFRET) have characterized the kinetics of the translocation events required for the CrPV-IRES to deliver an aminoacyl-tRNA to the P site of the ribosome (*Petrov et al., 2016*; *Zhang et al., 2016*). Slow movements of the CrPV-IRES compared with canonical translocation of tRNAs are evident from this data and explain the capturing of a late-stage intermediate of translocation with eEF2 in our dataset (*Figure 2A-C*). Our cryoEM reconstruction reveals that the conformation of the IRES in this intermediate state is similar to the conformation reported for the single translocated state (*Muhs et al., 2015*), with the SL-IV and SL-V detached from the 40S and exposed to the solvent. PKI is in an intermediate position between the P and E sites of the small subunit as well as the aminoacyl-tRNA is in an intermediate position between the A and P site of the 40S (*Figure 2B*). Domain IV of eEF2 occupies the A site of the small ribosomal subunit. This configuration is maintained by a distinctive swiveled configuration of the 40S head, resembling one of the late stages recently reported for the first translocation event (*Figure 2C*) (*Abeyrathne et al., 2016*).

The most populated class of particles represent a double translocated CrPV-IRES with aminoacyl-tRNA in a canonical configuration in the P site and eRF1* in the A site (*Figure 3*). Both aminoacyl-

**Table 1.** Data collection, model refinement and validation statistics.

**Data collection**

| Voltage (KV) | 300 | |
|---|---|---|
| Defocus range (μm) | −0.5/−3 | |
| Pixel size (Å/pixel) | 1.08 | |
| Electron dose (e⁻/Å²) | 64 | |
| Images collected | 16,303 | |
| Model Refinement | | |
| | CrPV-IRES/eRF1* | CrPV-IRES/eEF2 |
| Program/Protocol | Refmac5/Reciprocal space | Phenix/Real space |
| Resolution: | | |
| FSC 0.143 (Å) | 3.2 | 4.75 |
| Used in refinement (Å) | 3.8 | 7 |
| Map sharpening (Å) | −91.86 | −113.6 |
| Average B-factors (Å) | 169.58 | 394.12 |
| R.m.s deviations: | | |
| Bonds (Å) | 0.011 | 0.0033 |
| Angles (°) | 1.58 | 0.83 |
| Validation | | |
| Molprobity score | 2.81 | 1.50 |
| Clashcore, all atoms | 5.74 | 3.86 |
| Favored rotamers (%) | 88.51 | 99.7 |
| Ramachandran plot: | | |
| Outliers (%) | 3 | 0.02 |
| Favored (%) | 83.23 | 95.3 |

DOI: https://doi.org/10.7554/eLife.34062.004

tRNA and eRF1* populate conformations recently described, with the characteristic bent of the mRNA at the stop codon (*Figure 3—figure supplement 1D*; *Brown et al., 2015*). The small subunit in this reconstruction is in a non-rotated configuration and the 40S head is not tilted or swiveled (*Figure 3*). CrPV-IRES has undergone a conformational change that mainly affects PKI, but also the relative orientation of PKII and PKIII. In the pre-translocated as well as in the single translocated conformation of the IRES, PKII and PKIII interact by a network of non-covalent interactions involving sugar–sugar stacking interactions as well as A-minor interactions (*Figure 4—figure supplement 1*, (*Murray et al., 2016*; *Pfingsten et al., 2006*). This compact configuration adopted due to the physical proximity of PKII and PKIII seems to be a requirement for the initial binding to the ribosome as well as for the first translocation event (*Murray et al., 2016*; *Muhs et al., 2015*). In the state described here, these interactions are no longer established and a pronounced gap could be observed between both pseudoknots (*Figure 4B*, bottom right and *Figure 4—figure supplement 1*).

Upon back-swiveling of the 40S following eEF2 departure, the aminoacyl-tRNA is placed in its final canonical position in the P site (*Figure 3B*). This event triggers the disassembly of the CrPV-IRES PKI. Although the mRNA-like part remains placed in the E site of the 40S, the ASL-like segment experiences a pronounced displacement to occupy the E site of the 60S, now mimicking the acceptor stem of a canonical E site tRNA (*Figure 3B C*). The L1.1 part of CrPV-IRES remains attached to the L1 stalk along this process whose position relative to the 60S is similar to the one described in the complex with eEF2 and a non-hydrolyzable GTP analog or after the first translocation (*Figure 4A*)(*Murray et al., 2016*; *Muhs et al., 2015*). The back-swiveling of the 40S head upon eEF2 departure is also involved in a new relative orientation of PKII and PKIII (*Figure 3B C*). The swiveled configuration (*Figure 4B*, left), SL-IV and SL-V, components of PKII, are exposed to the solvent, in a

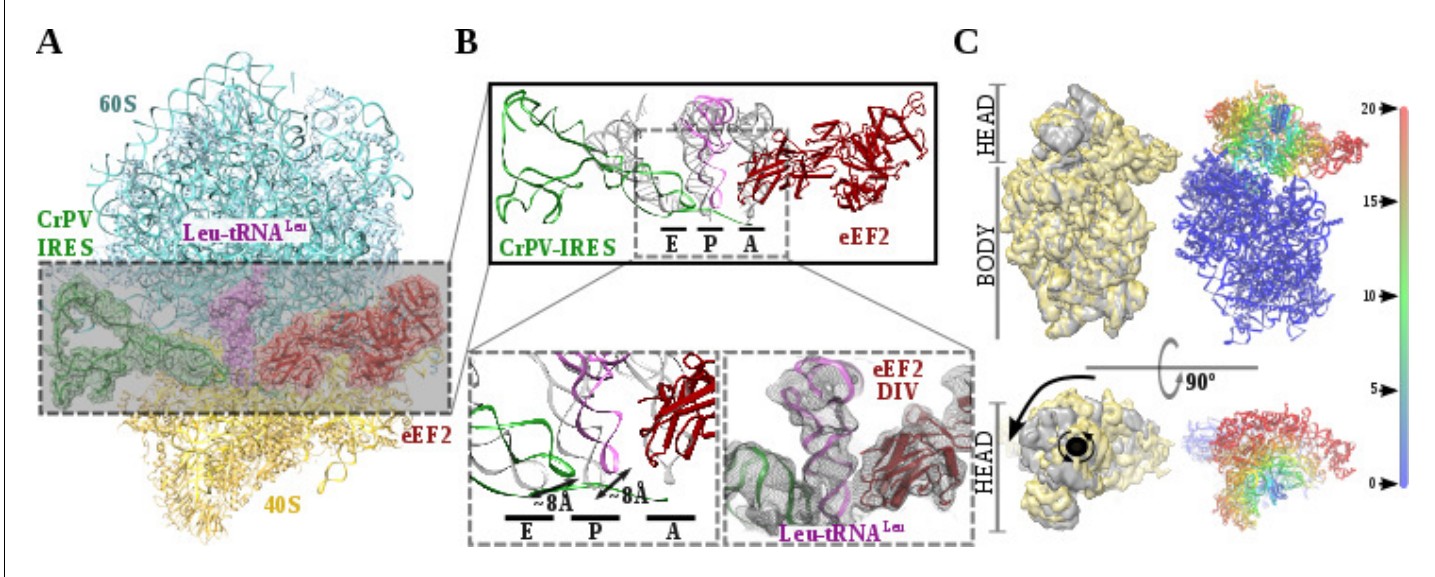

**Figure 2.** Structure of a double translocated CrPV-IRES intermediate with eEF2. (**A**) Overview of a mammalian ribosome with double translocated CrPV-IRES (green), aminoacyl-tRNA (purple) and eEF2 (red). (**B**) Top, detailed view of the ribosomal sites E, P and A in the structure with a double translocated CrPV-IRES with eEF2. Canonical tRNAs (from PDB ID 4V5C) are depicted as semi-transparent grey cartoons. Bottom left, zoomed view highlighting the displacement of both aminoacyl-tRNA and CrPV-IRES PKI from canonical positions. Domain IV of eEF2 occupies the A site. Bottom right, final experimental densities for the ribosome ligands described at 7 Å. (**C**) On the left, two views of the experimental density for the 40S of the two double translocated reconstructions where it can be appreciated the swiveled configuration of the 40S head in the complex with eEF2 (grey). For comparative purposes, the map for the eRF1* containing complex (yellow) has been low pass filtered to a similar resolution as the eEF2 containing complex (grey). Right, atomic refined model colored according to the root-mean squared displacement (RMSD) with red and blue indicating the highest and lowest values, respectively (in Ångstroms).

DOI: https://doi.org/10.7554/eLife.34062.005

The following figure supplement is available for figure 2:

**Figure supplement 1.** Fourier Shell Correlation curves and local resolution estimation for the 80S/CrPV-IRES/Leu-tRNA$^{Leu}$/eEF2 complex.

DOI: https://doi.org/10.7554/eLife.34062.006

similar position described for the single translocated CrPV-IRES (*Muhs et al., 2015*). The eukaryotic specific protein eS25, a key element of the small subunit involved in early recruitment of the CrPV-IRES as well as in the positioning of the IRES in the pre-translocation stage (*Schüler et al., 2006*), is not interacting with the IRES (*Figure 4B*, left arrow). Upon back swiveling of the 40S head, a new interaction is established between the CrPV-IRES and eS25 (*Figure 4B* right,arrow and (*C*) not involving SL-V like in the pre-translocated complex. The α-helix of eS25 comprising residues 52 to 65 could be observed in interacting distance with a helical segment of the CrPV-IRES formed by residues 6123–6127 and 6159–6164 (*Figure 4C*, top right). This new interaction stabilizes the CrPV-IRES in a distinctive conformation, with PKII and PKIII assembled but with a wider relative orientation (*Figure 4C*) and the PKI disassembled with residues 6175 to 6200 (*Figure 3C*)corresponding to the ASL mimicking part of PKI, populating a space corresponding to the acceptor stem of a canonical E site tRNA (*Figure 4C*, bottom right and *Video 1*).

## Discussion

The structures provide a structural view of the archetypical CrPV-IRES in the final stage of initiation, after transitioning through the ribosome. Combining the structures with published biochemical and smFRET data allows us to propose a comprehensive working model for how the CrPV-IRES (and type IV IRES in general) recruits, manipulates and redirects host ribosomes for the synthesis of its own proteins (*Figure 5*). As suggested by classic cross-linking experiments (*Pestova et al., 2004*), smFRET data (*Petrov et al., 2016*) and cryoEM reconstructions (*Spahn et al., 2004*; *Murray et al., 2016*), CrPV-IRES initially assembles a binary 80S/CrPV-IRES complex by either directly recruiting empty 80S or by a step-wise pathway in which CrPV-IRES first recruits the 40S subunit and then the

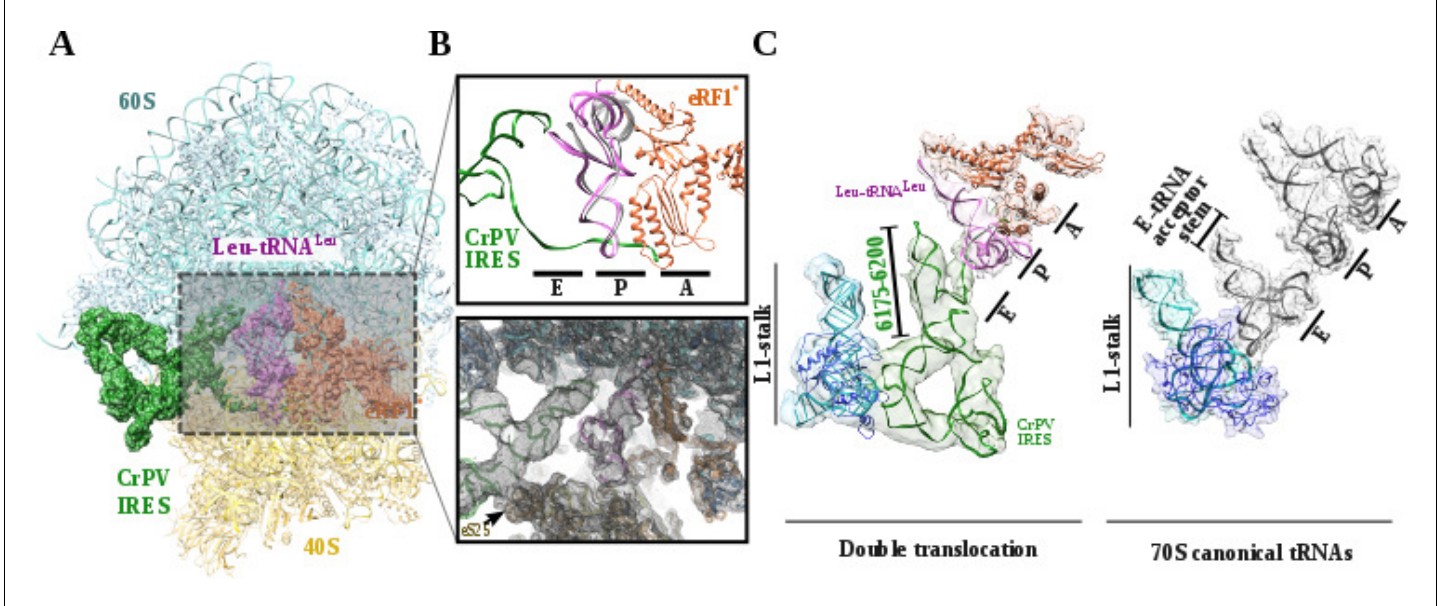

**Figure 3.** Structure of a double translocated CrPV-IRES. (**A**) Overview of a mammalian ribosome with double translocated CrPV-IRES (green), aminoacyl-tRNA (purple) and eRF1* (orange). (**B**) Top, detailed view of the ribosomal E, P and A sites with a canonical configuration for the aminoacyl-tRNA in the P site, eRF1* in the A site and a disassembled CrPV-IRES PK I in the E site (green). Canonical P site tRNA (from PDB ID 4V5C) is depicted as semi-transparent grey cartoon. Bottom, large field of view of the final unsharpened map obtained for this reconstruction focused on the area described. (**C**) The position of the three ligands in the double translocated complex with eRF1* in comparison with canonical tRNAs. The L1 stalk is depicted in cyan.

DOI: https://doi.org/10.7554/eLife.34062.007

The following figure supplement is available for figure 3:

**Figure supplement 1.** Fourier Shell Correlation curves and local resolution estimation for the 80S/CrPV-IRES/Leu-tRNA^Leu/eRF1* complex.

DOI: https://doi.org/10.7554/eLife.34062.008

60S subunit (*Figure 5*, bottom left). Once the binary 80S/CrPV-IRES is assembled, the 40S oscillates between rotated and non-rotated states, with PKI inserted in the A site and minimum changes in the overall conformation of the IRES (*Fernández et al., 2014*; *Koh et al., 2014*). These movements are coupled to oscillations of the L1 stalk. The rotated state is the substrate of eEF2, which, in its GTP-bound form, induces an additional rotation of the small subunit and additional displacement of the L1 stalk, to facilitate the translocation of the PKI from the A to the P site (*Figure 5A*, top left). Back rotation and back swiveling of the 40S, combined with ribosome-induced GTP hydrolysis by eEF2 results in the first translocation event of the CrPV-IRES, positioning PKI in the P site, mimicking a translocated, canonical aminoacyl-tRNA. This intermediate is unstable and prone to back-translocation (*Muhs et al., 2015*), unless a cognate aminoacyl-tRNA, delivered to the ribosome in complex with eEF1A and GTP, captures the frame in the A site of the ribosome (*Petrov et al., 2016*). Formation of this complex is a rate-limiting step in this kinetically driven process (*Petrov et al., 2016*). In the single translocated IRES state, SL-IV and SL-V, which are initially attached to the ribosome, are solvent exposed, the PKI occupy the P site of the 40S and an amino-acyl-tRNA occupies the A site. It is reasonable to assume this state will oscillate between rotated and non-rotated configurations of the small subunit as a canonical pre-translocation complex with tRNAs (*Budkevich et al., 2011*). The second translocation step is required to place the first amino-acyl-tRNA in the P site and thus finish the initiation phase of translation (*Figure 5A*, bottom right). At this stage, CrPV-IRES translocation should be coupled with the movement of the aminoacyl-tRNA occupying the A site, and this seems to happen with a conformation of the IRES similar to the one reported for the first translocation (*Muhs et al., 2015*). This conformation is maintained until the very last moment as the intermediate captured here with eEF2 presents a back-rotated configuration of the 40S (*Abeyrathne et al., 2016*). However, a pronounced swiveling of the 40S head is in place, probably induced by the presence of eEF2 (*Abeyrathne et al., 2016*). Once eEF2 leaves, the back-swiveling movement of the 40S head triggers a dramatic conformational change in the CrPV-IRES:

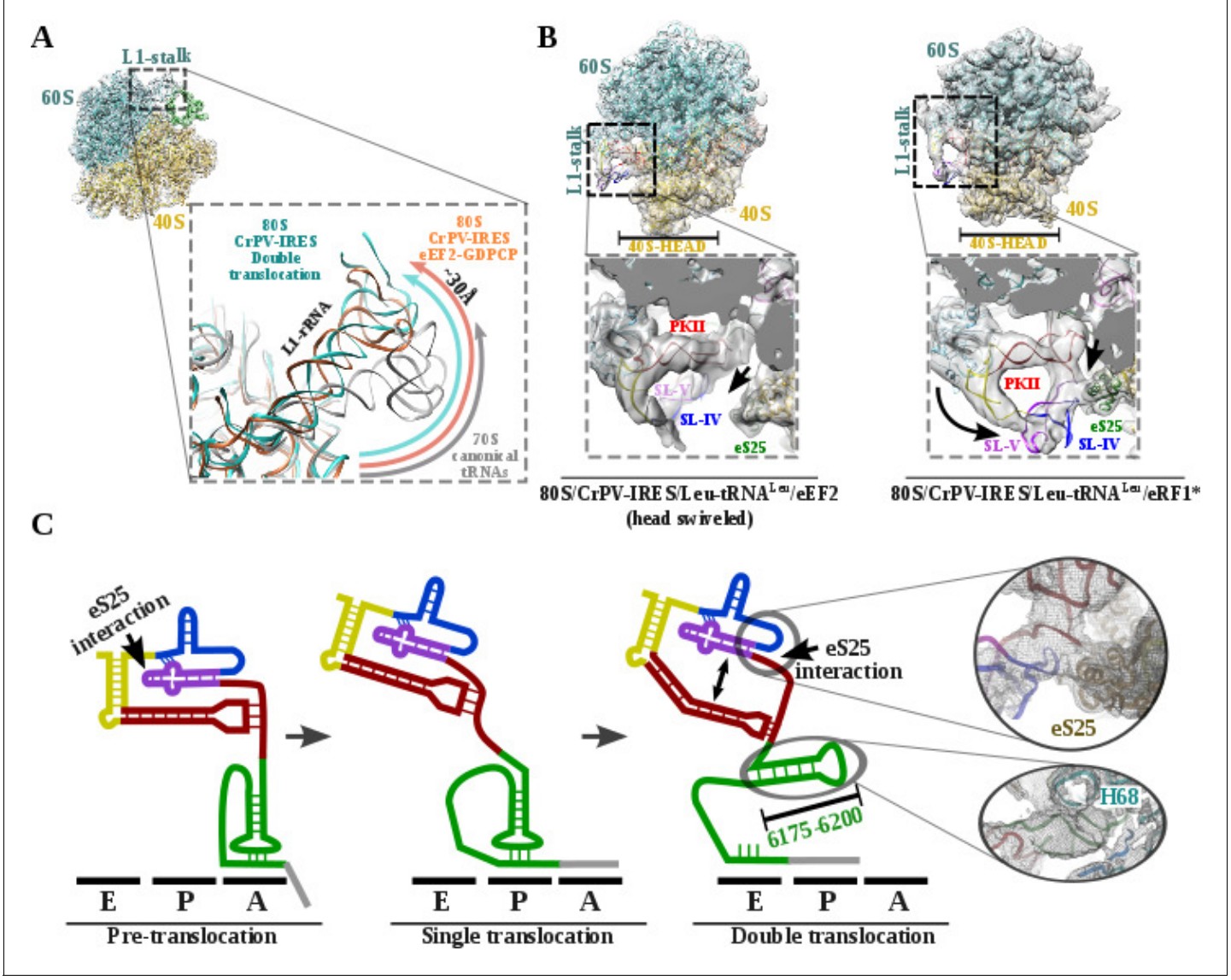

**Figure 4.** L1 stalk position and conformational change on double translocated CrPV-IRES. (A) Top left, overview of the double translocated ribosome complex with eRF1* with the L1 stalk region highlighted. Main view, L1 stalk in the double translocated complex (cyan) is displaced from the position acquired in a complex with canonical tRNAs (grey) with a magnitude of approximately 30 Å. This displacement is similar to the one reported for the pre-translocated complex with eEF2 and a non-hydrolyzable GTP analog (orange)(*Murray et al., 2016*). (B) Conformational transition observed in CrPV-IRES upon back-swiveling of the 40S head. Left, due to a swiveled 40S head configuration in the double translocated complex with eEF2, SL-IV and SL-V remain solvent exposed, as in the single translocated complex (*Muhs et al., 2015*), and detached from the ribosomal protein eS25 (green). Right, once the head of the 40S relocates to its non-swiveled position, CrPV-IRES acquires a new conformation involving a new interaction with eS25 (green). (C) Scheme showing the secondary structure of CrPV-IRES in the pre-translocated state (left), after a single translocation (center) and after the double translocation (right). Arrows indicate the repositioning of PKII and PKIII as well as the new interaction established with ribosomal protein eS25. On the right, close up views of the final unsharped map obtained for this reconstruction for the regions indicated by circles.

DOI: https://doi.org/10.7554/eLife.34062.009

The following figure supplement is available for figure 4:

**Figure supplement 1.** Structural transition observed in CrPV-IRESupon double translocation.

DOI: https://doi.org/10.7554/eLife.34062.010

PKI is disassembled resulting in the ASL-like segment relocating to mimic the acceptor stem of a canonical E site tRNA. The mRNA-like element of the disassembled PKI remains in the E site of the 40S. These conformational changes in the PKI of the CrPV-IRES upon back swiveling are combined with a reconfiguration of the relative positioning of PKII and PKIII. This new conformation is stabilized

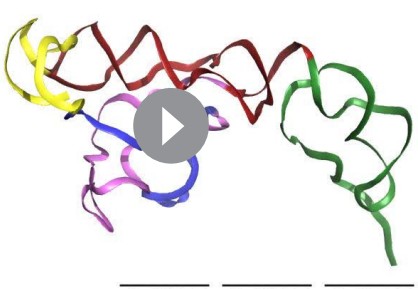

**E-SITE   P-SITE   A-SITE**

**Video 1.** Conformational changes experienced by the CrPV-IRES along its movement through the ribosome. CrPV-IRES binds initially to the ribosome inserting the PKI (green) in the A site (PDBID 5IT9, [*Murray et al., 2016*]). After a first translocation, PKI is placed in the P site (PDBID 4D61, *Muhs et al., 2015*) and a second translocation event induces its disassembly (this work). Molecular transitions have been approximated by a linear morph using Chimera (*Pettersen et al., 2004*). DOI: https://doi.org/10.7554/eLife.34062.011

by a newly reported IRES/40S interaction with the ribosomal protein eS25, which is also involved in the early recruitment of the IRES to the 40S (*Murray et al., 2016*).

The conformational change described here for the CrPV-IRES following translocation through the ribosome unexpectedly resembles the transition observed for the Hepatitis C Virus (HCV) IRES upon aminoacyl-tRNA delivery to the P site (*Figure 6*)(*Yamamoto et al., 2014*; *Yamamoto et al., 2015*). The HCV-IRES belongs to a different class of IRES, due to its requirement of some canonical factors to initiate translation (*Filbin and Kieft, 2009*; *Yamamoto et al., 2017*). It also interacts with the ribosome in a different manner (*Quade et al., 2015*). However, a large stem (*Figure 6*, domain II, blue) reaches the E site of the 40S and is maintained base paired with the mRNA-like part of this IRES by a tilted configuration of the 40S head (*Figure 6B*) (*Yamamoto et al., 2015*).Upon delivery of initiator tRNA to the P site, the head recovers its non-tilted configuration resulting in the repositioning of the domain II to occupy a similar space as the CrPV IRES in the E site of the 60S (*Figure 6*, right).

Therefore, to assemble translationally competent ribosomes, distantly related IRESs have converged on a similar mechanism to regulate the placement of the first aminoacyl-tRNA in the P site of the ribosome, by resembling endogenous tRNA states.

## Materials and methods

### Plasmids

Expression vector for His-tagged eRF1*(AGQ mutant) (*Frolova et al., 1999*) and transcription vector for Leu-tRNA have been previously described (*Pisarev et al., 2010*). Transcription vector for CrPV-Leu-STOP was constructed inserting a T7 promoter sequence upstream of CrPV IGR IRES sequence followed by the two first coding triplets and an EcoRI site, using pUC19 as a scaffold vector. Site-directed mutagenesis was employed to change the first coding triplet to CUU encoding leucine and the second coding triplet to a stop (UAG) codon, rendering the CrPV-Leu-STOP construct. CrPV-Leu-STOP RNA and Leu-tRNA were transcribed using T7 RNA polymerase.

### Purification of translation components and aminoacylation of Leu-tRNA

Native 40S and 60S subunits, eEF2, rabbit aminoacyl-tRNA synthetases (*Alkalaeva et al., 2006*), and eEF1A (*Carvalho et al., 1984*)were prepared as previously described. Recombinant eRF1* was purified according to a previously described protocol (*Alkalaeva et al., 2006*). In vitro transcribed Leu-tRNA was aminoacylated with leucine in the presence of rabbit aminoacyl-tRNA synthetases as previously described (*Pisarev et al., 2010*).

### Assembly of ribosomal complexes

To reconstitute different ribosomal complexes, we incubated 1.8 pmol 40S ribosomal subunits with 2 pmol CrPV-Leu-STOP RNA in a 20 µl reaction mixture containing buffer A (20 mM Tris-HCl, pH 7.5, 100 mM KCl, 2.5 mM MgCl2, 0.1 mM EDTA, 1 mM DTT) with 0.4 mM GTP for 5 min at 37. Then, the reaction mixture was supplemented with 2.5 pmol 60S ribosomal subunits and additionally incubated for 5 min at 37. Next, we added 10 pmol eEF1A, 3 pmol eEF2, and 0.4 Leu, and incubated for 5 min at 37. Finally, ribosomal complexes were incubated with 20 pmol eRF1 (AGQ) for 5

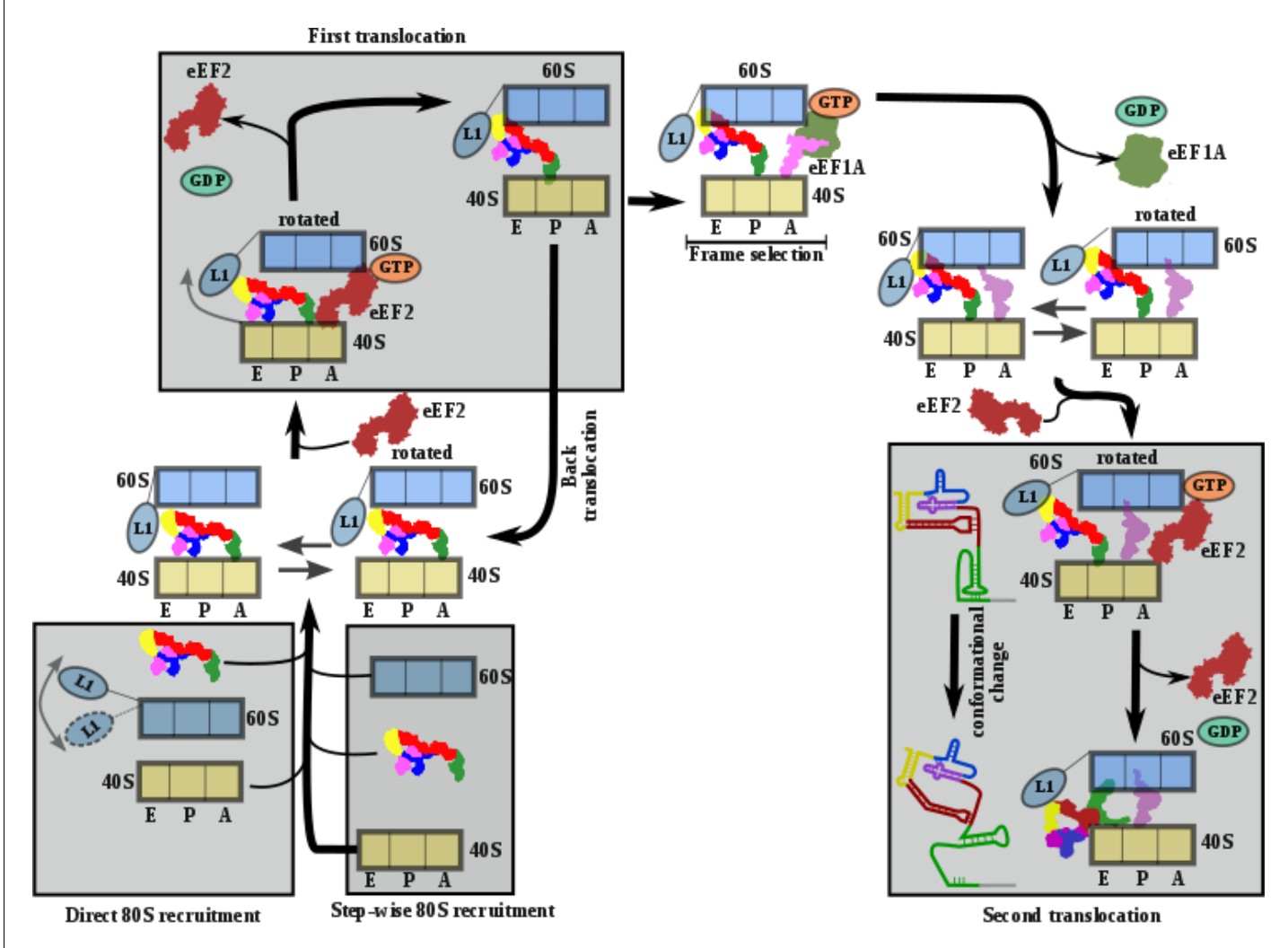

**Figure 5.** Comprehensive model describing CrPV-IRES strategy to hijack host ribosomes. Bottom left, CrPV-IRES can directly recruit 80S to assemble a binary 80S/CrPV-IRES complex which, in its pre-translocation state, oscillates between rotated and non-rotated configurations of the 40S. However, a step-wise pre-translocation complex formation involving an initial interaction with the 40S, followed by recruitment of 60S, is more efficient and is favored. Top left, first translocation event involving the displacement of CrPV-IRES PKI from the A site in order for the first aminoacyl-tRNA to be delivered as a ternary complex with eEF1A and GTP. In the absence of an A site ligand this state is unstable and prone to back translocation (*Muhs et al., 2015*). According to smFRET studies, the frame is not defined until the first condon/anticodon interaction is established (*Petrov et al., 2016*). Top right, presumably a single translocated complex with A site aminoacyl-tRNA alternates between rotated and non-rotated configurations of the 40S as a *bonafide* pre-translocation complex with two tRNAs. Bottom right, binding of eEF2 in its GTP form assists in the translocation of CrPV-IRES and the first aminoacyl-tRNA which is achieved through a conformational change in the CrPV-IRES involving the disassembling of the PKI and reorientation of PKII and PKIII.
DOI: https://doi.org/10.7554/eLife.34062.012

min at 37. We analyzed the assembled ribosomal complexes via a toe-printing assay essentially as described (*Pestova and Hellen, 2005*).

## CryoEM sample preparation and data acquisition

Aliquots of 3 μl of assembled ribosomal complexes at concentration of 80–100 nM were incubated for 30 s on glow-discharged holey gold grids (UltrAuFoil R1.2/1.3 (*Russo and Passmore, 2016*)), on which a home-made continuous carbon film (estimated to be 50Åthick) had previously been deposited. Grids were blotted for 2.5 s and flash cooled in liquid ethane using an FEI Vitrobot. Grids were transferred to an FEI Titan Krios microscope equipped with an energy filter (slits aperture 20 eV) and

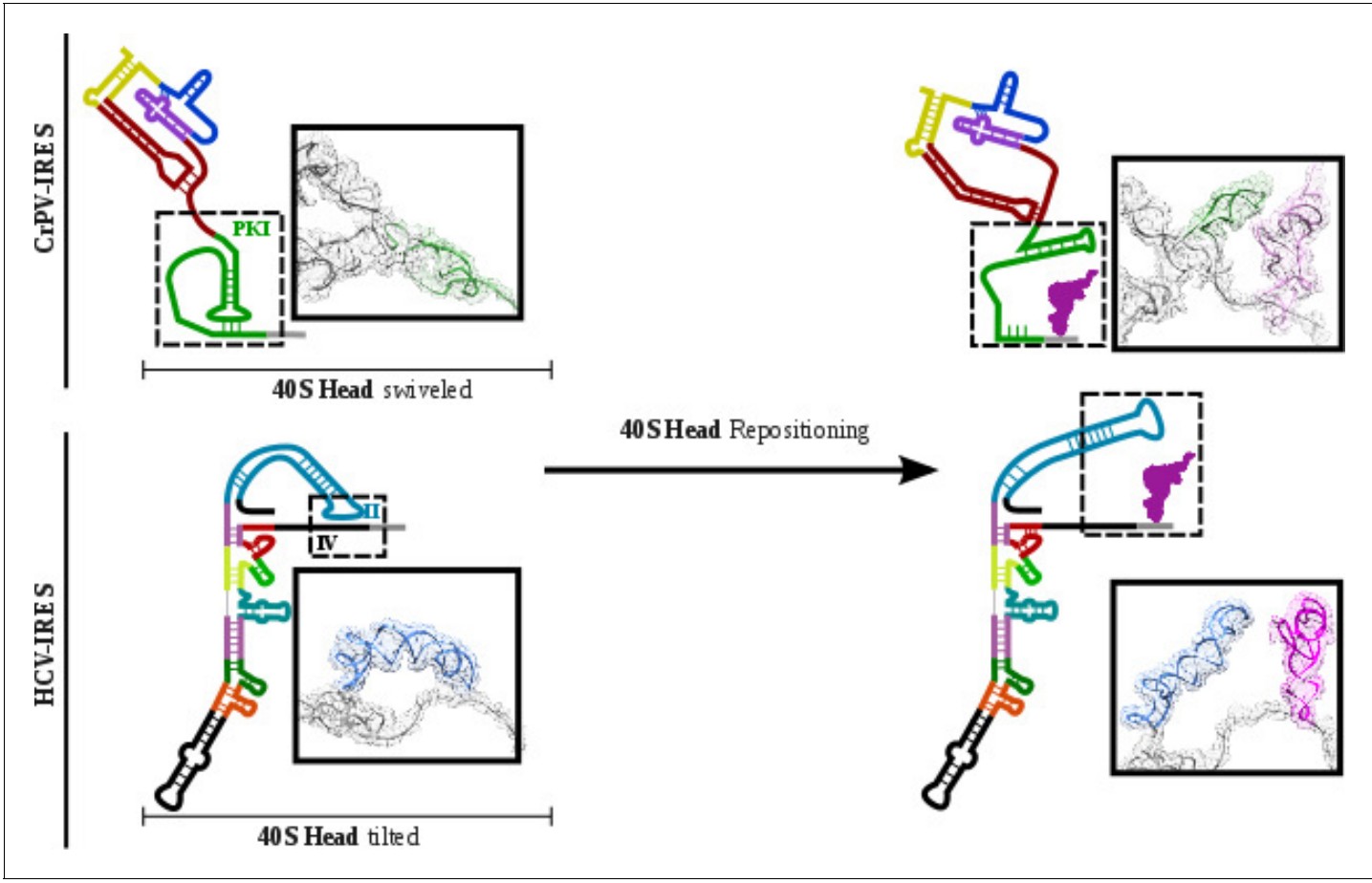

**Figure 6.** CrPV-IRES and HCV-IRES experiments a similar structural transition upon first aminoacyl-tRNA delivery to the ribosomal P site. The structural transition experienced by the CrPV-IRES upon delivery of the first aminoacyl-tRNA to the P site (top) is similar to the one described for the HCV IRES (bottom). CrPV-IRES PKI remains assembled and in the vicinity of the E site of the 40S due to a swiveled configuration of the 40S head. The HCV-IRES maintains a similar internal interaction of domain II by a tilted configuration of the 40S head. In case of both IRESs, a reconfiguration involving back-positioning of the 40S head plus the placement of a structural element of the IRES in the vicinity of the 60S E site, facilitates the delivery of the first aminoacyl-tRNA to the P site of the ribosome, finalizing the initiation stage of translation.

DOI: https://doi.org/10.7554/eLife.34062.013

a GatanK2 detector operated at 300 kV. Data were recorded in counting mode at a magnification of 130,000 corresponding to a calibrated pixel size of 1.08 Å. Defocus values ranged from 1.6 to 3.6 μm. Images were recorded in automatic mode using the Leginon (*Carragher et al., 2000*) software and frames were aligned with Motioncor2 (*Zheng et al., 2017*) and checked on the fly using APPION (*Lander et al., 2009*).

## Image processing and structure determination

Contrast transfer function parameters were estimated using GCTF (*Zhang, 2016*) and particle picking was performed using GAUTOMACH without the use of templates and with a diameter value of 260 pixels. All 2D and 3D classifications and refinements were performed using RELION (*Scheres, 2012*). An initial 2D classification with a four times binned dataset identified all ribosome particles. A second 2D classification step with two times binned data was employed to separate 80S from 40S particles. A consensus reconstruction with all 80S particles was computed using the AutoRefine tool of RELION whose resulting map was used to build a mask containing the inter-subunit space and the L1 stalk. 3D classification with signal subtraction using the previously described mask and a T value of 10 allowed for the identification of several population of ligands inside the mask, namely empty ribosomes, pre-translocated CrPV IRES and double translocated

populations with aminoacyl-tRNA and eEF2 or eRF1*. Final refinements with unbinned data for the classes selected yielded high resolution maps with density features in agreement with the reported resolution. Local resolution was computed with RESMAP (*Kucukelbir et al., 2014*).

### Model building and refinement

Models for the mammalian ribosome, Leu-tRNA$^{Leu}$, eEF2 and eRF1* were docked into the maps using CHIMERA (*Pettersen et al., 2004*) and COOT (*Emsley and Cowtan, 2004*) was used to manually adjust the L1 stalk and rebuild CrPV IRES using our previous model as initial step. An initial round of refinement was performed in Phenix using real space refinement with secondary structure restrains (*Adams et al., 2011*). A final step of reciprocal-space refinement using REFMAC was performed (*Murshudov et al., 1997*) for the eRF1* complex. The fit of the model to the map density was quantified using FSCaverage and Cref.

## Acknowledgements

We acknowledge Bob Grassucci for technical assistance in data acquisition, Harry Kao for computing and Alan Brown for critical reading of the manuscript. Some of this work was performed at the Simons Electron Microscopy Center and National Resource for Automated Molecular Microscopy located at the New York Structural Biology Center, supported by grants from the Simons Foundation (SF349247), NYSTAR, and the NIH National Institute of General Medical Sciences (GM103310). We are grateful to Ed Eng, Bill Rice, Laura Kim, Anchi Chen, Clint Potter and Bridget Carragher for support at all stages of this project. Density maps have been deposited at the EMDB with accession codes EMD-7834 for the eRF1* containing complex and EMD-7836 for the eEF2 containing complex. Atomic coordinates have been deposited in the PDB with accession numbers 6D9O and 6D9J. VPP and AVP are funded by National Institutes of Health [GM097014 to AVP] grant.

## Additional information

### Funding

| Funder | Grant reference number | Author |
|---|---|---|
| National Institutes of Health | GM097014 | Andrey V Pisarev |

The funders had no role in study design, data collection and interpretation, or the decision to submit the work for publication.

### Author contributions

Vera P Pisareva, Conceptualization, Supervision, Investigation, Writing—original draft, Writing—review and editing; Andrey V Pisarev, Investigation; Israel S Fernández, Supervision, Investigation

### Author ORCIDs

Israel S Fernández [ID] https://orcid.org/0000-0001-7218-1603

### Decision letter and Author response

Decision letter https://doi.org/10.7554/eLife.34062.028
Author response https://doi.org/10.7554/eLife.34062.029

## Additional files

### Supplementary files

• Transparent reporting form
DOI: https://doi.org/10.7554/eLife.34062.014

## Data availability

Density maps and coordinates have been deposited at the EMDB with accession codes EMD-7834 and 6D9O for the eRF1* containing complex and EMD-7836 and 6D9J for the eEF2 containing complex. Atomic coordinates have been deposited in the PDB with accession numbers 6D9O and 6D9J.

The following datasets were generated:

| Author(s) | Year | Dataset title | Dataset URL | Database, license, and accessibility information |
| --- | --- | --- | --- | --- |
| Fernández IS | 2018 | Density maps and coordinates for eRF1* containing complex | https://www.ebi.ac.uk/pdbe/emdb/EMD-7834 | Publicly available at the Electron Microscopy Data Bank (accession no: EMD-7834) |
| Fernández IS | 2018 | Density maps and coordinates for eRF1* containing complex | https://www.ebi.ac.uk/pdbe/emdb/EMD-6D9O | Publicly available at the Electron Microscopy Data Bank (accession no: EMD-6D9O) |
| Fernández IS | 2018 | Density maps and coordinates for eEF2 containing complex | https://www.ebi.ac.uk/pdbe/emdb/EMD-7836 | Publicly available at the Electron Microscopy Data Bank (accession no: EMD-7836) |
| Fernández IS | 2018 | Density maps and coordinates for eEF2 containing complex | https://www.ebi.ac.uk/pdbe/emdb/EMD-6D9J | Publicly available at the Electron Microscopy Data Bank (accession no: EMD-6D9J) |
| Fernández IS | 2018 | Atomic coordinates | http://www.rcsb.org/pdb/search/structidSearch.do?structureId=6D9O | Publicly available at the RCSB Protein Data Bank (accession no: 6D9O) |
| Fernández IS | 2018 | Atomic coordinates | http://www.rcsb.org/pdb/search/structidSearch.do?structureId=6D9J | Publicly available at the RCSB Protein Data Bank (accession no: 6D9J) |

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
