## [Decision Letter]

Thank you for submitting your article "Dual tRNA mimicry in the Cricket Paralysis Virus IRES uncovers an unexpected similarity with the Hepatitis C Virus IRES" for consideration by *eLife*. Your article has been favorably evaluated by John Kuriyan (Senior Editor) and three reviewers, one of whom is a member of our Board of Reviewing Editors. The reviewers have opted to remain anonymous.

The reviewers have discussed the reviews with one another and the Reviewing Editor has drafted this decision to help you prepare a revised submission.

Summary:

The CrPV IRES mechanism is unusual in that it adopts an RNA structure containing overlapping pseudoknots to recruit ribosomes directly and initiate translation without factors or initiator Met-tRNA. Structural studies using cryo-EM and x-ray crystallography have shown that the IRES occupies the E, P and A sites of the tRNA to manipulate the ribosome. Recent structures include a single translocated ribosome on the IRES as it moves from the initial A site to P site. Here, Fernandez et al. reconstitute ribosome translocation on the IRES to capture a structural view of a double translocated ribosome on the IRES by cryo-EM. Remarkably, the pseudoknot I (PKI) domain, which mimics an anticodon stem loop (ASL) of a tRNA and initially occupies the A site, undergoes a dramatic rearrangement in the double translocated complex. The PKI is disrupted and now rearranges such that the ASL domain resembles a tRNA acceptor stem loop in the E site. The authors provide a comparison to the HCV IRES-ribosome complex, where the domain II of the HCV IRES and the ASL of the CrPV IRES occupies the same tRNA acceptor stem space on the ribosome, thus revealing a similar strategy in manipulating the ribosome. In sum, the finding is surprising and highlights the amazing gymnastics of this IRES. The authors build on the model of IRES-ribosome translation and reveal a shared mechanism with a divergent HCV IRES. In general, the manuscript is clearly written in places but can be improved by providing more details and rationale in the experimental design in order to reach out to readers outside the IRES field.

Essential revisions:

1) The paper is aimed at a specialist audience, and is very difficult to read for people outside of the ribosome and IRES fields. The paper is also overly brief, and can use expansion to more fully explain points as well as to justify the conclusions. In particular:

- More description is needed of how the double translocated ribosome complex is reconstituted. Why use eRF1*? Can complexes be resolved without this? More details and rationale in this experiment is required, especially for the non-expert audience.

- There is a new interaction of eS25 with the IRES in the double translocated ribosome – this is quite vague. Are there any insights into the element (seems like the shoulder) of the IRES that is interacting? If so, maybe mutate that region to see effects on IRES translation and/or trap a double-translocated ribosome? Further proof of the eS25 interaction with this region would support the structural analysis.

- PKII and PKIII re-orient and become wider apart – the authors should explain more about the interactions that keep PKII and PKIII together (ex. A-minor interactions) and how these are disrupted in the final translocated step. Figure 4C should include these.

- The description of the new state (with the ASL-like part of PKI mimicking the E-tRNA acceptor arm) lacks detail and mechanistic insight. The major question is whether the interpretation of the map in this region is unambiguous. The following changes are required to address this question:

a) To illustrate the fit of PKI into the map, show a larger close-up view than in current figures (they are shown in tiny panels such as Figure 3C) to demonstrate the RNA hairpin features in the map (helical twist, backbone if visible etc.).

b) In canonical 70S-tRNA and 80S-tRNA complexes, the acceptor arm of E-tRNA is stabilized by interactions with the large subunit rRNA (including backbone-backbone packing with helices of 23S/25S/28S rRNA). Describe whether the PKI hairpin contacts the same elements of 28S rRNA.

c) Describe how the PKI hairpin interacts with L1.1 part of the IRES, which to some extent mimics the tRNA elbow? Does the map allow to say whether there is similarity to the structure of tRNA, in which the acceptor stem is coaxially stacked on the T stem?

- The fit and interpretation of the structure depend on the quality of the structural model, which is not reported in this manuscript (protein/RNA quality). The low local resolution of the IRES region makes misinterpretation possible. For example, the model of tRNA in Figure 6 (upper right box) looks distorted – is it due to over-refinement? To address this criticism, the authors should add (a) overall model statistics and (b) local IRES model statistics.

---

## [Author Response]

Essential revisions:1) The paper is aimed at a specialist audience, and is very difficult to read for people outside of the ribosome and IRES fields.

We have expanded the Introduction with two detailed paragraphs explaining briefly but accurately two main aspects of canonical translation which are relevant for understanding the biology of the CrPV IRES. The first added paragraph (Introduction, first paragraph) discusses the mechanism of translation initiation and how the initiator aminoacyl-tRNA is delivered to the ribosome in a canonical initiation. The second added paragraph (Introduction, fourth paragraph) is centered on canonical translocation and what is known on how the mobile ribosomal components are exploited to productively contribute to the movement of tRNAs and mRNA through the ribosome. We think non-specialist readers will find enough information in these two paragraphs to understand the results described in the manuscript.

The paper is also overly brief, and can use expansion to more fully explain points as well as to justify the conclusions. In particular:- More description is needed of how the double translocated ribosome complex is reconstituted. Why use eRF1*? Can complexes be resolved without this? More details and rationale in this experiment is required, especially for the non-expert audience.

We have added the following paragraph:

“We reasoned this ambiguity could be explained by the absence of an A site ligand, what would allow partial back translocation of the IRES even in the presence of a translocated aminoacyl-tRNA in the P site. […] Given the conformation of the P site tRNA is completely compatible with that of a translating ribosome (Figure 3), we believe the addition of eRF1* in the present sample does not significantly affect the conformation of the CrPV IRES.”

- There is a new interaction of eS25 with the IRES in the double translocated ribosome – this is quite vague. Are there any insights into the element (seems like the shoulder) of the IRES that is interacting? If so, maybe mutate that region to see effects on IRES translation and/or trap a double-translocated ribosome? Further proof of the eS25 interaction with this region would support the structural analysis.

The ribosomal protein eS25 is essential for initial binding of CrPV IRES to the small subunit as well as for inducing the rotational states in the context of the 80S critical for the first translocation. In our map of the double translocated CrPV IRES we found a new contact with this protein and we have added new panels with the experimental density in Figure 3 (panel B, bottom), Figure 4 (panels B right and C right, top). We have also added the following sentence:

“The α-helix of eS25 comprising residues 52 to 65 could be observed in interacting distance with a helical segment of the CrPV formed by residues 6123-6127 and 6159-6164”.

These IRES residues as well as eS25 are implicated in earlier binding events of the IRES to the ribosome, what would preclude a mutational analysis as it would be impossible to distinguish the effects impacting the initial binding from the translocation effect.

- PKII and PKIII re-orient and become wider apart – the authors should explain more about the interactions that keep PKII and PKIII together (ex. A-minor interactions) and how these are disrupted in the final translocated step. Figure 4C should include these.

We have added a new supplementary figure (Figure 4—figure supplement 1) as well as a video (Video 1) showing the confrontational transition experienced by the CrPV IRES from a pre-translocation state to a double translocated state. Details on how PKII and PKIII are co-stacked by a network of non-covalent interactions as well as the direction of the movement affecting mainly PKIII are illustrated (Figure 4—figure supplement 1). We have also included in the manuscript these sentences:

“In the pre-translocated as well as in the single translocated conformation of the IRES, PKII and PKIII interact by a network of non-covalent interactions involving sugar-sugar stacking interactions as well as A-minor interactions (Figure 4—figure supplement 1,). […] In the state described here, these interactions are no longer established and a pronounced gap could be observed between both pseudoknots (Figure 4B), bottom right.”

- The description of the new state (with the ASL-like part of PKI mimicking the E-tRNA acceptor arm) lacks detail and mechanistic insight. The major question is whether the interpretation of the map in this region is unambiguous. The following changes are required to address this question:a) To illustrate the fit of PKI into the map, show a larger close-up view than in current figures (they are shown in tiny panels such as Figure 3C) to demonstrate the RNA hairpin features in the map (helical twist, backbone if visible etc.).

We have added a panel in Figure 3B (bottom) with a wide field of view of the final model and experimental density in an orientation where it can be noted the fit of the model to the map.

b) In canonical 70S-tRNA and 80S-tRNA complexes, the acceptor arm of E-tRNA is stabilized by interactions with the large subunit rRNA (including backbone-backbone packing with helices of 23S/25S/28S rRNA). Describe whether the PKI hairpin contacts the same elements of 28S rRNA.

We have included a panel in Figure 4C (bottom right) with a top view of the model where it can be noted a similar placement of the disassembled ASL-like element of the CrPV IRES and the acceptor arm of a deacylated tRNA at the E site of the 60S. The helix 68 of the 28S rRNA, which is described to mediate interactions with the acceptor stem of a deacylated tRNA, is also indicated.

c) Describe how the PKI hairpin interacts with L1.1 part of the IRES, which to some extent mimics the tRNA elbow? Does the map allow to say whether there is similarity to the structure of tRNA, in which the acceptor stem is coaxially stacked on the T stem?

We respectfully disagree with the analysis of the reviewer at this point as not the PKI nor the ASL-like element of the CrPV IRES are ever in contact with the L1.1 region of the IRES. Additionally, the L1.1 segment of the IRES do not mimic the elbow of a tRNA in the present reconstruction or in any other. The interaction of the L1.1 region is established with the L1-stalk at the initial recruitment of the large subunit and is maintained remarkably similar as the IRES transit through the different tRNA binding sites of the ribosome. We can confirm with the present study and is illustrated in Figure 4A, that this L1-stalk/L1.1 interaction is maintained even after two translocation steps of the IRES.

- The fit and interpretation of the structure depend on the quality of the structural model, which is not reported in this manuscript (protein/RNA quality). The low local resolution of the IRES region makes misinterpretation possible. For example, the model of tRNA in Figure 6 (upper right box) looks distorted – is it due to over-refinement? To address this criticism, the authors should add (a) overall model statistics and (b) local IRES model statistics.

We have included a table with relevant information of data collection, model refinement and validation statistics. In regard to the specific point of the Figure 6, we would like to remind the reviewer the tRNA illustrated in this panel is from the final model presented in this work which is a Leu-tRNA^Leu.^ In contrast to all other tRNAs containing 4-5 nucleotides in the variable-arm (V-arm), tRNA^Leu^ has a long V-arm structure comprising 14 nucleotides, which is directly oriented towards the reader in this figure and thus it could be interpreted as a non-usual tRNA view.